# Insertion Hot Spots of *DIRS1* Retrotransposon and Chromosomal Diversifications among the Antarctic Teleosts Nototheniidae

**DOI:** 10.3390/ijms20030701

**Published:** 2019-02-06

**Authors:** Juliette Auvinet, Paula Graça, Laura Ghigliotti, Eva Pisano, Agnès Dettaï, Catherine Ozouf-Costaz, Dominique Higuet

**Affiliations:** 1Laboratoire Evolution Paris Seine, Sorbonne Université, CNRS, Univ Antilles, Institut de Biologie Paris Seine (IBPS), F-75005 Paris, France; paula.graca@sorbonne-universite.fr (P.G.); catherine.ozouf@orange.fr (C.O.-C.); 2Istituto per lo Studio degli Impatti Antropici e la Sostenibilità in Ambiente Marino (IAS), National Research Council (CNR), 16149 Genoa, Italy; laura.ghigliotti@cnr.it (L.G.); eva.pisano@ge.ismar.cnr.it (E.P.); 3Institut de Systématique, Evolution, Biodiversité (ISYEB), Museum National d’Histoire Naturelle, CNRS, Sorbonne Université, EPHE, 57, rue Cuvier, 75005 Paris, France; adettai@mnhn.fr

**Keywords:** Nototheniidae, chromosomal rearrangements, species radiation, retrotransposons, FISH, *DIRS1*, insertion hot spots

## Abstract

By their faculty to transpose, transposable elements are known to play a key role in eukaryote genomes, impacting both their structuration and remodeling. Their integration in targeted sites may lead to recombination mechanisms involved in chromosomal rearrangements. The Antarctic fish family Nototheniidae went through several waves of species radiations. It is a suitable model to study transposable element (TE)-mediated mechanisms associated to genome and chromosomal diversifications. After the characterization of *Gypsy* (*GyNoto*), *Copia* (*CoNoto*), and *DIRS1* (*YNoto*) retrotransposons in the genomes of Nototheniidae (diversity, distribution, conservation), we focused on their chromosome location with an emphasis on the three identified nototheniid radiations (the *Trematomus*, the plunderfishes, and the icefishes). The strong intrafamily TE conservation and wide distribution across species of the whole family suggest an ancestral acquisition with potential secondary losses in some lineages. *GyNoto* and *CoNoto* (including *Hydra* and *GalEa* clades) mostly produced interspersed signals along chromosomal arms. On the contrary, insertion hot spots accumulating in localized regions (mainly next to centromeric and pericentromeric regions) highlighted the potential role of *YNoto* in chromosomal diversifications as facilitator of the fusions which occurred in many nototheniid lineages, but not of the fissions.

## 1. Introduction

The roles of transposable elements (TEs) in species diversification has emerged from various sources [1,2,3] but still remain only partially understood. Multiple studies have established a correlation between TE bursts in various lineages and speciation events [4,5,6,7]. However, potential mediation of TEs in species divergence, including via chromosomal rearrangements, remains still a highly debated topic [8,9]. Characterization of TE genomic landscapes and chromosomal insertion patterns in various taxonomic groups are essential to better understand their putative involvement in genome evolution.

The family of Antarctic teleosts Nototheniidae is the major group of the Austral Ocean fish fauna [10,11,12]. They represent 77% of the species diversity and 91% of the fish biomass of the Southern Ocean [10,13]. In addition to the acquisition of antifreeze glycoproteins [14,15], the predominance of the Nototheniidae is due to several waves of rapid diversifications on the Antarctic continental shelf [16,17,18,19]. Among them, three radiations: the *Trematomus* (including *Indonotothenia cyanobrancha* and *Pagothenia borchgrevinki* [18,20,21]), the Artedidraconinae (plunderfish) and the Channichthyinae (white-blooded icefish) (Figure 1) have all the criteria of a marine species flock [12,13,22,23]. Those radiations are consecutive to a changing environmental context with series of glacial and warming periods associated with dynamic changes in the icecap and leading to habitat fragmentations or free circulation on the continental shelf [24,25,26,27]. This propelled the Nototheniidae to a high level of specific, ecological and morphological diversity [13,23]. Moreover, in some nototheniid groups (sub-families and genera), specific diversity was accompanied by significant chromosomal variability [28,29,30], rare among other marine teleosts [31,32]. Assuming a plesiomorphic state of 2*n* = 48 small acrocentrics [33,34,35], the observed intra (locality, sex differentiation) [28,29,30,36,37,38] and inter-specific chromosomal flexibility [28,29,30] result from multiple rearrangements. These gave rise to very small acrocentrics, large metacentrics or submetacentrics [29,33,39] that occurred during nototheniid diversifications [20,29,33,36,38,40].

Like for numerous other eukaryotic taxa [41,42,43], transposable elements (TEs) have been found widely distributed and in various proportions in fish genomes [6,44,45,46,47]. Their abundance represents less than 2% in the compact genomes of pufferfishes *Takifugu rubripes* and *Tetraodon nigroviridis* [48,49,50]. It varies between 14 and 28% of the genome of the spotted gar *Lepisosteus oculatus*, the stickelback *Gasterosteus aculeatus*, the Atlantic cod *Gadus morhua*, the platyfish *Xiphophorus maculatus*, the tilapia *Oreochromis niloticus*, and the medaka *Oryzias latipes* [46,47], and represents up to 55% of the zebrafish *Danio rerio* genome [51]. Compared to other Vertebrates, TE diversity is generally higher in teleost fish genomes [6,46,52]. While retrotransposons (class I TEs) represent the majority of the elements in cichlids and in *D. rerio* [51,53], DNA transposons (class II TEs) are dominant in *Takifugu rubripes* and in *Lepisosteus oculatus* genomes [50,54]. In Antarctic teleost Nototheniidae, TEs contribute to about 12.5% of the black rockcod *Notothenia coriiceps* and blackfin icefish *Chaenocephalus aceratus* genomes, with all eukaryote TE super-families represented [55,56].

Several TEs like the class II *Tc1*-like, *Helitron2* (*Helinoto*) and class I *Rex*1/3, *Gypsy* (*GyNoto*), *Copia* (*CoNoto*) and *DIRS1* (*YNoto*) have already been studied in the genome of different nototheniid representatives [20,57,58,59]. The chromosomal location descriptions mentioned multiple TE copies mostly dispersed along chromosomes (arms and extremities). Three of them (the *Tc1*-like, the *Rex*3 and the *DIRS1* TEs) revealed hot spots of insertions in heterochromatic regions [20,58,59]. In the *Trematomus* group at least some of them (the *DIRS1* elements) might have facilitated chromosomal rearrangements, mostly fusions [20].

In the present study, we focused on three class I TE super-families: the two long terminal repeats (LTR) type retrotransposons super-families *Gypsy* and *Copia*, and the Tyrosine recombinase (YR) type retrotransposon *DIRS1* [60,61,62]. Their diversity, distribution, conservation, as well as their location on chromosomes were investigated in various nototheniids (Figure 1), including species with karyotypes close to the plesiomorphic chromosomal number as well as species with rearranged karyotypes (more or less than 2*n* = 48). All the examined TEs were widely distributed and highly conserved, but only *DIRS1* elements (named *YNoto*) revealed a particular insertion pattern with strong accumulations in centromeric and pericentromeric chromosomal regions. In the changing environmental context characterizing the nototheniid species flocks and their karyotype diversification, our results support a role of *YNoto* elements as facilitators of the chromosomal fusions.

## 2. Results

### 2.1. Gypsy, Copia and DIRS1 Diversity in Nototheniid Genomes

#### 2.1.1. Identification and Distribution

TE families which were identified in *Trematomus* species [20]: *YNotoJ*, *V*, *R* and *B*, *GyNotoA*, *B*, *E*, *F*, *H*, *I* and *RT*, and *CoNotoB* were found to have a wide distribution in every examined nototheniid species (Table 1). The targeted search for these elements spanning the Reverse Transcriptase (RT), the RNAseH (RH) and/or the Integrase (Int) conserved domains resulted in the occasional identification of additional TE families. While investigation of their presence and distribution in the genomes of the other species was out of the scope of the present work, we included them into the analysis to complete the overview of TEs in nototheniid genomes. Three new families of *Gypsy* (*GyNotoC*, *G*, and *K*) were detected in *Notothenia angustata*, *Lepidonotothen nudifrons* and *Gobionotothen gibberifrons* respectively (Table 1).

The family *CoNotoA* previously identified in the genome of *Notothenia coriiceps* and *Dissostichus mawsoni* [20] belongs to the *GalEa* clade [64]. It was found in the Nototheniinae, the Dissostichinae, the Channichthyinae, the Gymnodraconinae, the Artedidraconinae, and in the Gobionototheniinae representatives except in *Paranotothenia magellanica*, *Chionodraco hamatus*, and *Histiodraco velifer* (Table 1). However, as amplification of cloned PCR products was used in this study, we assumed that a lack of identification of a given TE family in a species using this approach does not mean they are absent from their genome. Five other TE families (*CoNotoB*, *C*, *D*, *E* and *F*) belonging to the *Hydra* clade [65] were found. Their distribution was more patchy (Table 1).

#### 2.1.2. Sequence Proximity and TE Clustering

Although the clustering threshold for the delimitation of the TE families was set to 80% identity, we found an even higher sequence conservation across species from the same genus or sub-family inside every identified TE family. When taking into account every examined nototheniid group, this high conservation corresponds to 91% average nucleotide identity for the *YNoto* families, 93% or 94% for the fragments of *GyNoto* that span the RT/RH or the Int portions, and 96.5% for the *CoNoto* families (Table 2).

Surprisingly, this high degree of sequence identity was maintained at larger taxonomic scale for more distant species from the different sub-families. It can vary between 76 and 98.1% for the *YNoto*, 81.4 and 99% for the *GyNoto* -RT/RHportion, 77.6% and 99.1% for the *GyNoto* -Int portion, and between 61.7 and 98.8% for *CoNoto* families accross all nototheniid groups (Table 2, Appendix A).

Distance and maximum likelihood reconstructions positioned all the identified TE families of *YNoto*, *GyNoto* and *CoNoto* retrotransposons relative to one another. The same topologies were observed for distance and maximum likelihood reconstruction methods. As expected, each sequence from a same TE family clustered together to form monophyletic groups in the topologies of *YNoto*, *CoNoto*, and *GyNoto* (RT/RH or Int portions) (Figure 2A–D).

*YNotoJ* and *V* seemed more similar to each other than to the other sequences, as are *YNotoR* and *B* (Figure 2A). All TE families from the *Hydra* clade were more similar to each other than to the *CoNotoA* family belonging to the *GalEa* clade (Figure 2B). Inside the *Hydra* clade, the families *CoNotoC*, *D*, *E*, and *F* were more similar to each other than to *CoNotoB* (Figure 2B). In the same way, *GyNotoA* and *E* seemed more similar to each other than to the rest of the *Gypsy* sequences that span the RT/RH region (Figure 2C), and *GyNotoE* and *F* seemed more similar to each other than to the rest of the *Gypsy* sequences that span the Int region (Figure 2D).

The long branch separating the *GyNotoRT* and *I* from the other *Gypsy* families could be due to problems encountered in sequence alignments (Figure 2C,D and Appendix A), as well as nucleotide sequence divergence, also identified between the two *Copia* clades (Figure 2B and Appendix A).

The two additional *DIRS1* sequences (766 and 1418 available in the FishTEDB [66]) originating from the genome sequencing of *N. coriiceps* [67] clustered with the *YNotoJ* and the *YNotoR* families respectively (arrows in Figure 2A). The four additional sequences of *Gypsy* did not cluster with a *GyNoto* family identified in our analysis. The *Gypsy* sequence 864 was more similar to the *GyNotoB* and *D*, while sequences 99, 296 and 490 appeared to be new families (arrows in Figure 2C,D).

### 2.2. Gypsy, Copia and DIRS1 Locations on Nototheniid Chromosomes

Locations of TEs on the chromosomes of nototheniid species were investigated using fluorescent in-situ hybridization (FISH). *YNoto* (family *YNotoJ*), *GyNoto* (family *GyNotoA*), and *CoNoto* (families *CoNotoA* (*GalEa* clade) and *CoNotoB* (*Hydra* clade)) were hybridized on chromosome preparations from three nototheniid species with slight rearrangements in their karyotypes (the icefish *Chionodraco hamatus,* 2*n* = 47 (male)/48 (female), the plunderfish *Histiodraco velifer*, 2*n* = 46, and the ploughfish *Gymnodraco acuticeps*, 2*n* = 48) [28,37,68,69]. The probed TE families were chosen for their sequence size (insert > 1 kb) and their large distribution in the genomes of nototheniid species (Table 1, Appendix A).

Two major types of TE distribution patterns were identified (Figure 3A). Distribution pattern (1) is characterized by dense accumulation (hot spots of insertions) mainly next to centromeric and/or pericentromeric regions, and sometimes in intercalary or near telomeric positions. Distribution pattern (2) is defined by scattered, punctuated staining along chromosome arms. Combinations of these two patterns were also observed. The distributions (1) and/or (2) clearly depended upon the TE super-family, but not upon TE families [20].

The *GyNotoA*, *CoNotoA* and *CoNotoB* elements were mostly dispersed throughout nototheniid chromosome arms, producing a type 2 pattern. They formed multiple spots scattered all along the chromosomes including next to the centromeric and telomeric regions.

Contrary to the other TEs examined, hybridizations of the *CoNotoA* probe showed soft signals, not detected in every chromosome pair (for example not observed in the large sub-metacentric pair of *H. velifer*, Figure 3A(g), e.g., purple arrows). They are mostly localized in intercalary regions of acrocentric chromosomes in the three species. On the Y sex-differentiated chromosome of *C. hamatus*, we noticed one small band of *CoNotoA* near the centromere, and another band in the middle of the long arm (Figure 3A(c), e.g., red and white arrows).

The *CoNotoB* probe occasionally showed signals accumulated in pericentromeric regions of one or two pairs of acrocentric chromosomes, giving a type 1 + 2 distribution pattern (Figure 3A(d,l)).

The *YNotoJ* location was very distinct from the *GyNoto* and *CoNoto*, with clear hot spots of insertions giving a type 1 distribution pattern in all nototheniid species studied. It includes also the three additional *Trematomus* species: the spotted notothen *T. nicolai* (2*n* = 57 (male)/58 (female)), the emerald rockcod *T. bernacchii* (2*n* = 48), and the blunt scalyhead *T. eulepidotus* (2*n* = 24) [28,36,70]. They essentially accumulated in centromeric and pericentromeric regions (white arrows in Figure 3A,B), including the large metacentric or sub-metacentric pairs and the short acrocentric pairs (Figure 3A(a,e,i) and 3B(a–c)).

*YNoto* formed strong aggregations in centromeric regions and intercalary positions on the Y sex-differentiated chromosome of *C. hamatus* (Figure 3A(a), e.g., red and white arrows), and in centromeric and pericentromeric regions on the largest sub-metacentric pair of *H. velifer* (Figure 3A(e)), e.g., purple and white arrows).

Almost every chromosome pair was marked when using the *YNotoJ* probe (hot spots or soft signals). The number of unlabeled chromosomes increased when using the *CoNotoB* and *GyNotoA* probes. Last, the *CoNotoA* probe provided the weakest signal, labeling only few (five or six maximum) pairs with punctual spots (Figure 3A,B).

When focusing on chromosomal fusions, in addition to *C. hamatus* and *H. velifer*, we localized elements of the *YNotoJ* family on the chromosomes of two other nototheniids presenting highly rearranged karyotypes with massive fusions [29,36,39,71]: *T. eulepidotus* and *N. coriiceps* (respectively 2*n* = 24 and 22 large metacentrics or submetacentrics) (Figure 4). In the comparative analysis of these four species, the *YNotoJ* elements seemed systematically aggregated in centromeric regions (*T. eulepidotus*, except for the pair number 7 and *C. hamatus*) and in pericentromeric regions (*N. coriiceps* and *H. velifer*) of large metacentric or sub-metacentric pairs supposed to result from centric fusions [29,36,37,39]. Accumulation of *YNotoJ* was also detected at intercalary position on the longest arms of the largest pair (number 1) in *T. eulepidotus*, the pairs number 3 and 4 in *N. coriiceps*, and on the Y sex-differentiated chromosome in *C. hamatus*. All of these pairs may originate from successive rearrangements including a tandem fusion between a sub-metacentric and an acrocentric chromosome [33,37,38,71].

In the examined nototheniid species, *YNotoJ* was detected in the majority of chromosomal pairs forming pericentromeric accumulations (Figure 3 and Figure 4) and was found even systematically present in case of putative centric or tandem fusions (Figure 4). However, we noticed a higher proportion of unlabeled chromosomes in *T. nicolai* (2*n* = 57/58), especially in small pairs supposed to result from fission events (Figure 3B and Appendix A).

## 3. Discussion

### 3.1. TE Diversity

The TE families *YNotoJ*, *V*, *R* and *B*, *GyNotoA*, *B*, *E*, *F*, *H*, *I* and *RT*, and *CoNotoB* identified in the *Trematomus* genomes [20] were also widely distributed in the other examined nototheniid species (Table 1). The targeted research of these elements also led to the identification of new TE families, especially many new *GyNoto* elements. These are known to be very diversified in eukaryote genomes through their evolutionary strategy named “red queen theory” [72]. In the *CoNoto* elements, new families from the *Hydra* clade (*CoNotoC*, *D*, *E*, and *F*) have been detected.

We noticed a strong sequence conservation across species within the same TE family. Nucleotide identity percentages within a nototheniid genus or sub-family are comparable to those identified for the *Trematomus* genus [20]. The conservation at the scale of the whole nototheniid family remains very high (mean of 88 to 90% nucleotide identity across genera or sub-families) (Table 1, Appendix A) and equivalent to typical genic conservation [16,18,19]. This conservation suggests shared TE mobilization(s) and diversification(s) occurring before the episodes of rapid speciations in nototheniid lineages (23.9 My, Matschiner et al. [73]; 22.4 My, Near et al. [16]; 15 My, Colombo et al. [19]). Further alignments with TEs identified in other Eupercaria species [74] could give information about the timing of TE mobilization and diversification in those genomes. Preliminary results indicate proximity with some copies. These shared TEs would indicate their acquisitions in a common ancestor, before the separation between Antarctic and temperate species and the formation of the Antarctic convergence. However, better additional alignments with well assembled Eupercaria TE sequences and non-chimeric elements would be needed to further investigate the origin of the *YNoto*, *GyNoto* and *CoNoto* in nototheniid genomes and to relate it to the geographic and oceanographic events leading to the isolation from temperate species of this teleost group. Since speciation is not a discrete process and reproductive barriers take time to establish (evolutionary scale), we cannot exclude the influence of interspecific hybridizations in favoring transpositions as described in plants or insects in a genomic destabilization and re-organization context [75].

*GalEa* TEs have been detected neither in trematomine genomes, nor in *P. magellanica*, *C. hamatus*, *C. mawsoni*, or *H. velifer* (Table 1). Considering that this *Copia* clade is known to have a clade-dependant distribution and can be secondarily lost in entire taxonomic groups [76], our results suggest TE losses in several groups of Nototheniidae, possibly occurring several times in the distinct lineages Trematominae, Nototheniinae, Channichthyinae, Cygnodraconinae, and Artedidraconinae. However, we cannot yet conclude that the lack of detection of a given TE or family in a genome (“−“ in Table 1) means they are absent. For example, no *Copia* TEs have been found in the genome of *C. hamatus* by our targeted approach (cloning of PCR products amplified with specific designed primers for each identified family), but *CoNotoA* and *CoNotoB* have been visualized on chromosomes of the crocodile icefish by heterologous *FISH* mapping.

Even if FISH is not precise enough to estimate accurate TE copy numbers, we could compare the relative abundance between signals provided by different TEs. We generally observed fewer signals of *CoNotoA* than *CoNotoB* on chromosomes of every examined nototheniid species (*C. hamatus*, *H. velifer* and *G. acuticeps*). It would be interesting to map chromosomal locations of those *Copia* elements on chromosomes of other species in which they have been described to assess whether the predominance of the *Hydra* clade over to the *GalEa* clade is generalizable in other eukaryote lineages.

Although representing a step forward in the current knowledge, the results obtained by cloning are not exhaustive and the search for TE diversity in nototheniid genomes would certainly benefit from more resolutive NGS shotgun sequencing [67,77].

### 3.2. A Role of the DIRS1 in the Chromosomal Fusions within the Family Nototheniidae?

The *YNoto* insertion pattern is really distinct from the *GyNoto* and *CoNoto* locations on chromosomes of *T. bernacchii*, *T.eulepidotus*, *N. coriiceps*, *C. hamatus*, *H. velifer*, and *G. acuticeps*, like it was shown before for *T. pennellii*, *T. hansoni* and *D. mawsoni* [20]. *YNoto* systematically formed strong insertion hot spots in centromeric and pericentromeric regions in the case of chromosomes formed by hypothesized centric (centromere–centromere) fusions, and intercalary aggregations in case of potential tandem (centromere–telomere) fusions between two chromosomal segments (Figure 3 and Figure 4). Those regions might correspond to putative privileged breakpoints becoming junction points between fused chromosomal segments inherited from the 24 pairs of the last common ancestor of the family.

Through ectopic recombination mechanisms associated to double strand DNA breaks and cell reparation, TE insertions may participate in structural rearrangements [6,20,78,79]. Although the correlation between TE mobilization and chromosomal rearrangements in a species or a taxon has been repetitively mentioned in the literature [4,43,80,81,82,83,84], the causal link between TE mobilization and chromosomal diversification is not easy to demonstrate and remains a debated question [8,9]. In the context of the ongoing debate, our results on the very peculiar insertion pattern of *DIRS1* in putative chromosomal junction points provide support in favor of their possible involvement in the chromosomal rearrangements that occurred in the Nototheniidae. If we consider that *DIRS1* mobilization is certainly not the only factor driving rearrangement events in Nototheniidae, present evidence supports the hypothesis that they could have played a role as facilitators of the fusions observed in the whole family.

Models relating *DIRS1* and the chromosomal fusions that accompanied nototheniid diversifications should take into account the context of the oceanographic and environmental changes in the Austral Ocean from ending Eocene (habitat fragmentation, iceberg scouring, changing of the water level) [85,86,87] and its strong impact on the local biological systems. Such an environmental instability may have boosted *DIRS1* mobilizations in nototheniid genomes and the accumulation of numerous copies next to pericentromeric and centromeric regions in almost every chromosomal pair as well as near the telomeric regions for some of them. Double strand DNA breaks following each transposition event could have led to ectopic recombination for some chromosomal pairs, resulting in fused chromosomes that are around twice the size of the original acrocentrics. Centric fusions would have led to new metacentrics or submetacentrics while tandem fusions would have led to new large acrocentrics.

As an alternative hypothesis and in the same context of DNA instability, *DIRS1* TEs could have taken advantage of the DNA breakage to insert and accumulate in centromeric, pericentromeric, and sometimes in telomeric regions thanks to their “homing” properties [88]. However, in that case, what could have generated the DNA breaks instead of transpositions, especially in regions where recombination is restrained [89]? Under the hypothesis of TE opportunist insertions, we would expect a similar distribution pattern whether the chromosomes are fused or fissioned, as both were subjected to double strand DNA breaks. The results on *T. nicolai* (2*n* = 57/58) provide some interesting additional information. This species possesses the highest chromosomal number among Nototheniidae, with five small pairs thought to have originated from chromosomal fissions. The location of *YNoto* on chromosomes of *T. nicolai* revealed hot spots of insertions in centromeric and pericentromeric regions for the largest acrocentric pairs (probably inherited from the 2*n* = 48 acrocentrics ancestral nototheniid karyotype) and for a very limited number of small acrocentrics (Figure 3B). In contrast *YNoto* copies seem totally absent in most small acrocentrics (probably fissioned pairs) or can rarely form very little spots next to centromeric regions (Figure 3B and Appendix A). This result goes strongly against the *DIRS1* opportunist insertion alternative hypothesis whereas it well supports the hypothesis of a *DIRS1* mobilization older than the chromosomal breakage, in addition to restrict their putative role to nototheniid chromosomal fusions.

### 3.3. Toward an Evolutionary Scenario of Nototheniid Chromosomal Speciations

While their role in speciation remains controversial, chromosomal rearrangements are important in genome evolution [90,91,92,93]. They are considered as a major force accompanying species diversifications [2,3,83]. When transmitted and fixed within a population (homozygous state), chromosomal repatternings appear to play a major part in reproductive isolation process favoring diversifications [92,94,95]. This barrier to gene flow between individuals works through reduction of the fitness of the hybrids (heterozygous for the rearrangement) and/or recombination locking at fused sites where “speciation genes” might be located [93].

Given their wide distribution in nototheniid genomes, their high degree of intrafamily conservation, their high abundance compared to *Gypsy* and *Copia* TEs [20] and their conserved pattern of chromosomal insertion, *DIRS1* bursts could have occurred early during the recent nototheniid species diversifications. The timing of the mobilizations remains to be determined. They could have happened during warm periods, giving rise to a karyotypic polymorphism when nototheniids could circulate freely on the continental shelf (sympatric situation) [96]. *DIRS1* bursts could also have occurred during cold periods, with emergence of rearranged chromosomal pairs in nototheniids isolated in micro-refuges (allopatric situation) [96]. Rearrangements could then be fixed in the populations, *a fortiori* faster in the case of cold periods with small isolated population sizes [90,97,98].

The spectacular ecological and chromosomal diversity of the *Trematomus* radiation [20] stands in strong contrast with the other waves of nototheniid diversifications. Even if the timing of speciation events is really difficult to calibrate among the Nototheniidae due to the lack of fossils, knowledge of the environmental context during their diversification is crucial to improve our understanding of this group. Among the three nototheniid flocks, the *Trematomus* was the oldest one (with datations varying between −11 and −6 My ± 3.9 My whereas they were evaluated between −6.3 and −3.5 My ± 2.6 My for the channichthyids, and between −3 and −1.2 My ± 1.7 My for the much more recent artedidraconids) [16,19]. We can hypothesize that *DIRS1* TEs may have had more time to be mobilized in *Trematomus* genomes in addition to accumulate in specific chromosomal regions prone to centric or tandem fusions. Moreover, the trematomine radiation corresponds to a cooling period occurring during the mid–end Miocene [86,87]. Fixations of the fusions could have been much easier in those small isolated populations living in refuges dispersed all around the continental shelf [10,13]. The following warm period could have led to recolonization of vacant ecological niches and diversifications—specialization of lifestyles on a wide depth distribution [10,12,13,23,99], with for example the cryopelagic species *T. borchgrevinki* (2*n* = 45/46), *T. brachysoma* [96,100,101], and the deeper species *T. loennbergii* [99]. According to our results, *DIRS1* were also mobilized in artedidraconids and white-blooded icefish genomes. However, both radiations occurred more recently and were associated to the strong variability of temperatures that characterizes the Pliocene and Pleistocene ages. It might have reduced the probability of occurrence and fixation of the putative fusions.

## 4. Materials and Methods

### 4.1. Fish Specimens

Specimens of twenty nototheniid species (*Trematomus eulepidotus*, *T. bernacchii*, *T. nicolai*, *Lepidonotothen nudifrons*, *L. larseni*, *L. squamifrons*, *Patagonotothen guentheri*, *Pleuragramma antarctica*, *Notothenia coriiceps*, *N. angustata*, *N. rossii*, *Paranotothenia magellanica*, *Dissostichus mawsoni*, *Chionodraco hamatus*, *Cryodraco antarcticus*, *Champsocephalus gunnari*, *C. esox*, *Cygnodraco mawsoni*, *Gymnodraco acuticeps*, *Pogonophryne scotti*, *Histiodraco velifer*, *Gobionotothen gibberifrons*) were collected during French, Italian, US and international Antarctic and subantarctic groundfish survey programs. Fish specimens and their tissue and chromosome preparations are referenced in Appendix A, with corresponding campaigns and localities of capture. Nomenclature and classification of the SCAR atlas was adopted in this study [63].

### 4.2. Sample Collection and Preparation

#### 4.2.1. Tissues for DNA Analyses

Muscle samples or fin clips for DNA analyses were stored in 85% ethanol at −20 °C. DNA was prepared following the protocol of Winnepenninckx et al. [102].

#### 4.2.2. Chromosome Preparations

Mitotic chromosomes were obtained from head kidney and spleen. For most fish specimens, direct in vivo method, according to Doussau de Bazignan and Ozouf–Costaz (1985) [103] has been followed with few modifications: fish were treated with colchicine (0.3 mL per 100 g weight) during 10 to 15 h prior to sacrifice; mitotic cells were obtained from cephalic kidney and spleen, and hypotonized one hour at 0 °C prior to fixation in absolute ethanol/acetic acid 3:1. For specimens from Adelie–Land, chromosome preparations were obtained strictly following the cell culture protocol of Rey et al. [104]. Fixed cells were preserved at ‒20 °C. They were spread onto Superfrost slides (pre-cleaned with absolute ethanol containing 1% of 1 N HCl) that have subsequently been stored at −20 °C until the FISH step.

### 4.3. TE Amplification in Nototheniid Genomes

Characterization of the different TE families in the twenty-two nototheniid species were performed by PCR amplification using primers previously designed on TE in *Trematomus* species [20]. For the *DIRS1* elements (*YNotoJ*, *V*, *R*, *B*), the size of the amplification fragments was 1.25 kb and overlapped the RT/RH region. The *Gypsy* fragments overlapped the RH/Int regions (1.25 kb, *GyNotoA*, *B*, *D*, *E*, and *J*), only the Int region (0.6 kb, *GyNotoF*, *H*, and *I*) or the RT domain (0.6 kb, *GyNotoRT*). For *Copia* elements, fragment sizes were 0.95kb (*CoNotoA* (*GalEa* clade) or 1.38 kb (*CoNotoB*, *Hydra* clade)) and spanned the RT/RH region [20,64].

PCR was performed using 50 ng of genomic DNA, 2.5 U of Taq DNA polymerase (Promega) and 50 pmol of each degenerate primer in a final volume of 25 µL for 35 cycles (94 °C for 45 s, 50.2 °C for 1 min and 72 °C for 1 min). PCR products were visualized on 1% agarose gels. Fragments of the expected molecular weights were excised, purified with the Nucleospin Extraction kit (Macherey_Nagel, Düren, Germany), and cloned into the pGEM-T vector according to the supplier’s recommendations (Promega, Madison, WI, USA). Cloned fragments were sequenced in both directions (http://www.gatc-biotech.com).

Clustering to assemble the different families was performed using the BLASTClust toolkit v2.2.26 [105]. The criteria for inclusion of a fragment in a cluster (i.e., family) were ≥30% sequence coverage and ≥80% sequence identity.Phylogenetic analysis of nototheniid retrotransposons.

TEs identified in nototheniid species were used to run clustering and phylogenetic analysis. In order to equilibrate the nototheniid clades representation, three of the twelve *Trematomus* species examined for TE exploration were retained. This choice corresponds to *T. eulepidotus* [20], *T. bernacchii* and *T. nicolai* where TEs have been located on chromosomes in the present study. Multisequence alignments were performed with MAFFT v7 [106] and ambiguously aligned sites were removed using Gblocks [107] and BioEdit [108]. Neighbor-joining (NJ) trees were obtained using GENEIOUS v9.0.2 (http://www.geneious.com, [109]), implemented with the Jukes–Cantor genetic distance model. Maximum Likelihood (ML) reconstructions were obtained using RAxML 8.2.12 [110] and the evolution model GTRGAMMA. Support for individual clusters was evaluated using non-parametric bootstrapping [111] and 1000 bootstrap replicates. Topologies were presented as unrooted trees using GENEIOUS software.

### 4.4. FISH

#### 4.4.1. TE Probe Preparation

*DIRS1*, *Gypsy* and *Copia* clones from *C. hamatus*, *H. velifer*, *P. scotti* and *G. acuticeps* greater than 1 kb in length were used as probes for *FISH* experiments (Appendix A). *TE* probes were biotinylated by nick translation according to the manufacturer’s instructions (Roche Diagnostics, Mannheim, Germany). Each probe was dissolved at a final concentration of 25 ng/µL in high stringency hybridization buffer (65% formamide, 2 × SSC, 10% dextran sulfate (pH 7)).

#### 4.4.2. FISH with TE Probes

FISH was performed according to the protocol of Bonillo et al. [112], which is optimized for repetitive probes and multi-copy genes. Hybridization parameters, especially time of chromosome denaturation were adjusted to each species chromosome preparation [20].

#### 4.4.3. Image Acquisition and Karyotyping

FISH signals were detected using a Zeiss Axioplan microscope equipped with a cooled CCD camera (Coolsnap Photometrics, Tucson, AZ 85706, USA) and an XCite LED fluorescence light source. Karyotypes were processed using CytoVision 3.93.2/Genus karyotyping-FISH-imaging software for animal chromosomes (Leica Microsystems, Wetzlar, Germany). Ten to forty metaphase spreads/species for each probe were examined. The karyotype of *T. nicolai* was generated using the manually classification function of the CytoVision software.

### 4.5. Ethics Approval and Consent to Participate

Ethical approval for all procedures was granted by the ethics committee of the Ministère de l’Environnement and the French Polar Research Institute (Institut Paul Emile Victor–IPEV), which approved all our fieldwork. The experiments complied with the Code of Ethics of Animal Experimentation in the Antarctic sector.

### 4.6. Data Availability

The datasets (nototheniid TE sequences) supporting the conclusions of this article are available in the GenBank NCBI repository (BankIt2177835, refs MK330226 to MK330429).

## 5. Conclusions

This study explored in a wider sampling of nototheniid genomes the occurrence of three TE superfamilies *DIRS1* (*YNoto*), *Gypsy* (*GyNoto*) and *Copia* (*CoNoto*), previously characterized only within *Trematomus* species. This targeted research also led to the identification of new TE families, especially for *GyNoto* and *CoNoto* elements. We showed the wide distribution of these TEs among all species investigated, with a strong sequence conservation. This suggests shared TE mobilization(s) and diversification(s) occurring before the episodes of rapid speciations in the nototheniid lineages.

Chromosomal FISH mapping of these three TE superfamilies highlights their differential patterns of insertion. The *DIRS1* elements accumulated in insertion hot spots of the pericentromeric regions while *Gypsy* and *Copia* were dispersed throughout chromosome arms. Because of this particular pattern of insertion, *YNoto* could mediate nototheniid chromosomal diversifications like those first described in the genus *Trematomus* [20]. While this involvement seems only applicable to centric or more rarely to tandem fusions, it was observed in numerous nototheniid sub-families, including the barbled plunderfish (Artedidraconinae) and the crocodile icefish (Channichthyinae) radiations (species with 2*n* < 48). Conversely the *YNoto* do not appear to have any impact on chromosomal fissions, as shown in *T. nicolai* (2*n* > 48).

## Figures and Tables

**Figure 1 ijms-20-00701-f001:**
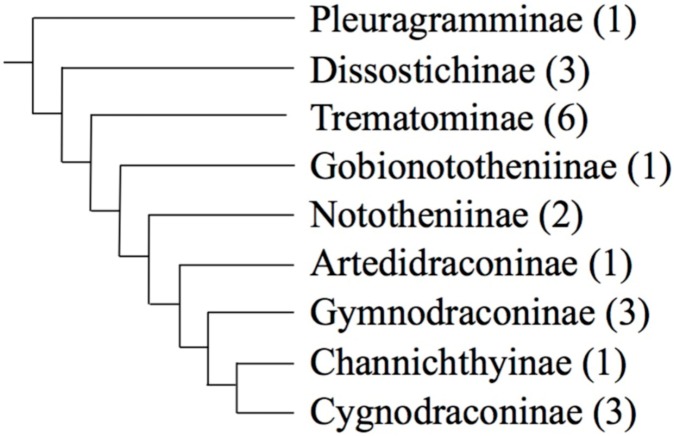
Phylogenetic relationships (cladogram) of the nototheniid sub-families presented in this study (as defined in [63]). Total number of genera for each nototheniid sub-families are indicated in parenthesis. Genera used in this study: *Pleuragramma* (Pleuragramminae), *Dissostichus* (Dissostichinae), *Trematomus*, *Lepidonotothen*, *Patagonotothen* (Trematominae), *Gobionotothen* (Gobionototheniinae), *Notothenia*, *Paranotothenia* (Nototheniinae), *Histiodraco*, *Pogonophryne* (Artedidraconinae), *Champsocephalus*, *Chionodraco*, *Cryodraco* (Channichthyinae), *Cygnodraco* (Cygnodraconinae), *Gymnodraco* (Gymnodraconinae).

**Figure 2 ijms-20-00701-f002:**
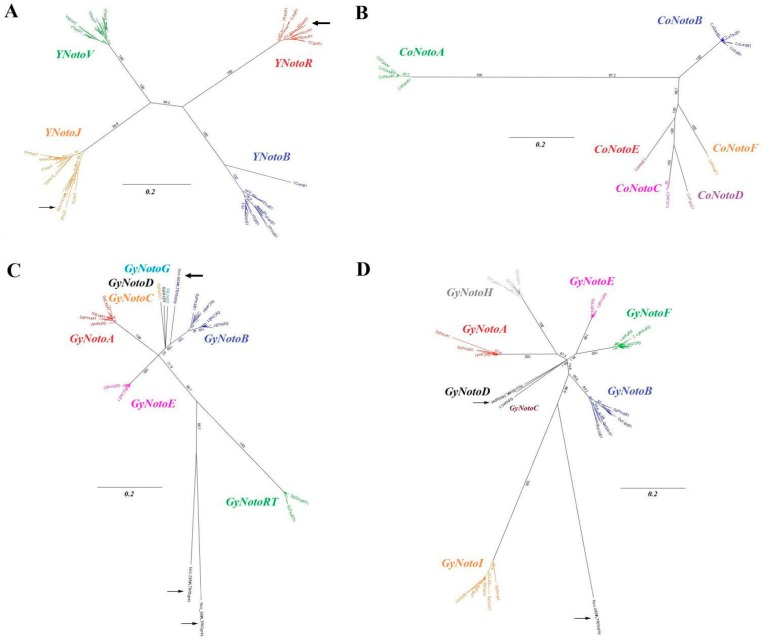
Neighbor joining (NJ) unrooted trees for (**A**) *YNoto*, (**B**) *CoNoto*, (**C**) *GyNoto* (RT/RH) and (**D**) *GyNoto* (Int) nucleotide sequences for the whole nototheniid transposable element (TE) datasets. Alignements used to construct these trees are presented in Appendix A. Analyses were run using the Jukes–Cantor model and no outgroup. Support for individual clusters was evaluated using non-parametric bootstrapping with 1000 replicates. Arrows show the additional *DIRS1* and *Gypsy* sequences originating from the *N. coriiceps* genome sequencing [67].

**Figure 3 ijms-20-00701-f003:**
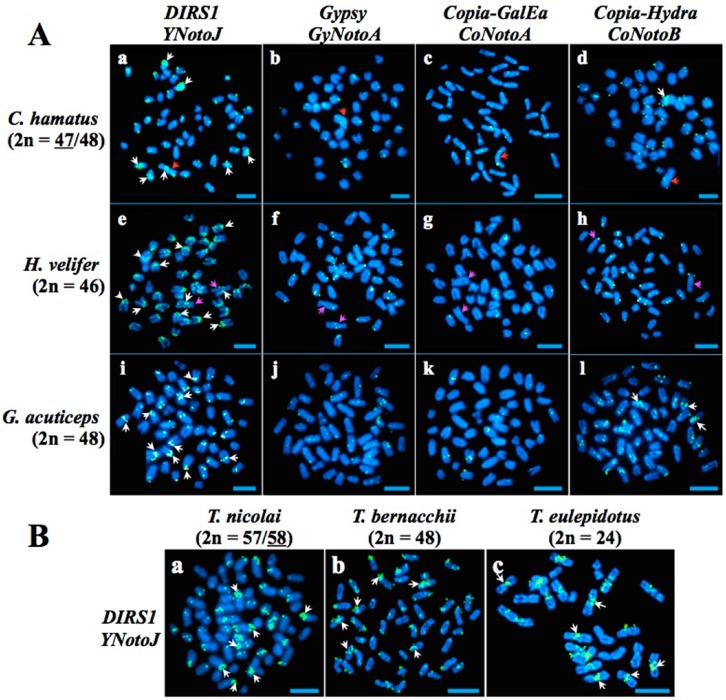
Fluorescent in-situ hybridization (FISH)-mapping of TEs on the chromosomes of six nototheniid species. (**A**) *YNoto*, *GyNoto* and *CoNoto* positioning in three non *Trematomus* Nototheniidae, (**B**) *YNoto* positioning in three *Trematomus* species. The chromosomal numbers of *C. hamatus* and *T. nicolai* are sex dependent. The number for the sex represented in the figure (male for *C. hamatus*, female for *T. nicolai*) is underlined. Each probe was labeled with biotin and bound probes were detected with incubation with Avidin-FITC (fluorescein, greenish spots). Probe characteristics are indicated in Appendix A. Chromosomal DNA was counterstained with 4′,6-diamidino-2-phenylindole (DAPI). One family from each retrotransposon superfamily is represented in this figure for *YNoto* (*YNotoJ*) and *GyNoto* elements (*GyNotoA*), and two families for *CoNoto* elements (*CoNotoA* (*GalEa* clade) and *CoNotoB* (*Hydra* clade)). Examples of TE distributions for pattern 1: a, e, i; pattern 2: j, h; and pattern 1 + 2: d, l. White arrows point examples of TE accumulations. Red arrows point the heteromorphic Y sex chromosome in *C. hamatus* and purple arrows indicate the largest sub-metacentric pair in *H. velifer*. Scale bars: 10 μm.

**Figure 4 ijms-20-00701-f004:**
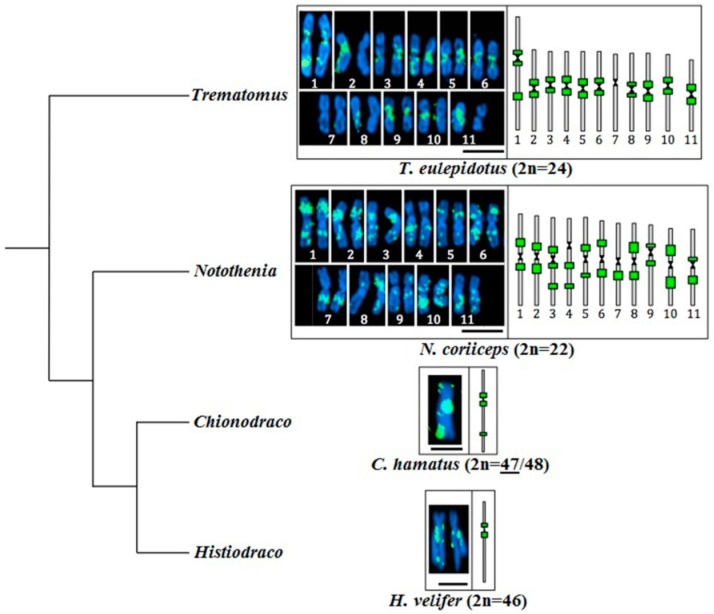
FISH patterns of *YNoto* on fused chromosomes of *T. eulepidotus*, *N. coriiceps*, *C. hamatus*, and *H. velifer*. The chromosomal number of *C. hamatus* is sex dependent. The number for the sex represented in the figure (male) is underlined. Diploid sets for DAPI captures and haploid sets for scheme representations are presented. The main *YNoto* insertion hot spots observed by FISH are represented by the green rectangles. Scale bars: 7 µm.

**Table 1 ijms-20-00701-t001:** Distributions of four families of *DIRS1* (*YNoto*), seven families of *Gypsy* (*GyNoto*) and six families of *Copia* (*CoNoto*) in the nototheniid examined genomes.

TE Super-Families	*DIRS1 (YNoto)*				*Gyspy* (*GyNoto*)							*Copia* (*CoNoto*)		
TE Families	*YB*	*YJ*	*YR*	*YV*	*GyA*	*GyB*	*GyE*	*GyF*	*GyH*	*GyI*	*GyRT*	*CoA*	*CoB*	Other
Species/TE Regions	*RT/RH*				*RH/Int*			*Int*			*RT*	*RT/RH*	*RT/RH*	
*PLEURAGRAMMINAE*														
*Pleuragramma antarctica*	+	+	+	+	−	+	−	+	+	+	+	−	+	
*DISSOSTICHINAE*														
*Dissostichus mawsoni **	+	+	+	+	+	+	+	+	+	+	+	+	+	
*TREMATOMINAE*														
*Trematomus bernacchii **	+	+	+	+	+	+	+	+	+	+	+	−	+	
*Trematomus Nicolai **	+	+	+	+	+	+	+	+	+	+	+	−	+	
*Trematomus eulepidotus **	+	+	+	+	+	+	+	+	+	+	+	−	+	
*Lepidonotothen nudifrons*	+	+	+	+	+	+	−	+	+	+	+	−	+	
*Lepidonotothen larseni*	+	+	+	+	+	+	+	+	+	+	+	−	+	
*Lepidonotothen squamifrons*	+	+	+	+	+	+	+	+	+	+	+	−	+	
*Patagonotothen guentheri*	+	+	+	+	+	+	+	+	+	+	+	−	−	*CoNotoD*
*GOBIONOTOTHENIINAE*														
*Gobionotothen gibberifrons*	+	+	+	+	+	+	−	+	+	+	+	+	−	*CoNotoC*
*NOTOTHENIINAE*														
*Notothenia coriiceps **	+	+	+	+	+	+	+	+	+	+	+	+	+	
*Notothenia angustata*	+	+	+	+	+	+	+	+	+	+	+	+	−	*CoNotoF*
*Notothenia rossii*	+	+	+	+	+	+	+	+	+	+	+	+	+	
*Paranotothenia magellanica*	+	+	+	+	+	+	+	+	+	+	+	−	−	*CoNotoF*
*ARTEDIDRACONINAE*														
*Pogonophryne scotti*	+	+	+	+	+	−	+	+	+	+	+	+	+	
*Histiodraco velifer*	+	+	+	+	+	+	+	+	+	+	+	−	−	*CoNotoE*
*GYMNODRACONINAE*														
*Gymnodraco acuticeps*	+	+	+	+	+	+	+	+	+	+	+	+	+	
*CHANNICHTHYINAE*														
*Chionodraco hamatus*	+	+	+	+	+	+	+	+	+	+	−	−	−	
*Cryodraco antarcticus*	+	+	+	+	+	+	+	+	+	+	−	+	−	*CoNotoC*
*Champsocephalus gunnari*	+	+	+	+	+	+	+	+	+	+	−	+	−	*CoNotoC*
*Champsocephalus esox*	+	+	+	+	+	+	+	+	+	+	−	+	−	*CoNotoC*
*CYGNODRACONINAE*														
*Cygnodraco mawsoni*	+	+	+	+	+	+	+	+	+	+	+	−	−	*CoNotoE*

RT/RH stands for the Reverse Transcriptase/RNAseH, RH/Int for RNAseH/Integrase and Int for Integrase conserved domains. +: transposable elements (TEs) tested and found in the genomes (at least one cloned sequence per species), −: TEs tested but not found using our approach. *: data from Auvinet et al. [20]. Nototheniid sub-families are indicated in uppercase, bold font, and underlined. Species in which *YNoto*, *GyNoto*, and *CoNoto* TEs have been located on chromosomes are presented in bold font.

**Table 2 ijms-20-00701-t002:** TE intrafamily conservation within and across nototheniid clades.

Clades	TEs	Trem	Pleu	Noto	Diss	Chan	Cygn	Gymn	Arte	Gobi
Trem	*YNoto*	93.2	86.9	87.1	85.9	87.8	88.5	89.3	88.2	89.0
	*GyNoto*	93.8	90.0	91.9	92.1	92.4	93.6	91.4	92.6	91.2
	*CoNoto*	96.0	95.3	95.8	92.2	NA	NA	97.1	88.4	NA
Pleu	*YNoto*		NA	85.1	83.0	85.9	86.3	90.0	86.4	85.5
	*GyNoto*		NA	88.5	84.5	89.0	95.0	82.0	84.1	96.3
	*CoNoto*		NA	95.8	92.1	NA	NA	96.5	88.2	NA
Noto	*YNoto*			87.4	88.9	86.4	88.5	87.4	89.7	89.5
	*GyNoto*			90.2	91.0	92.1	92.3	91.5	92.6	90.5
	*CoNoto*			96.4	92.6	NA	NA	96.9	89.7	NA
Diss	*YNoto*				NA	85.1	85.6	89.5	86.6	82.0
	*GyNoto*				NA	92.1	92.8	92.8	93.0	88.8
	*CoNoto*				NA	NA	NA	96.2	86.7	NA
Chan	*YNoto*					86.5	91.2	90.3	92.0	86.5
	*GyNoto*					92.7	93.7	92.6	93.5	91.3
	*CoNoto*					NA	NA	NA	NA	NA
Cygn	*YNoto*						NA	94.6	93.2	91.0
	*GyNoto*						NA	90.0	93.3	92.0
	*CoNoto*						NA	NA	NA	NA
Gymn	*YNoto*							NA	91.9	86.0
	*GyNoto*							NA	96.7	82.8
	*CoNoto*							NA	89.5	NA
Arte	*YNoto*								95.5	89.5
	*GyNoto*								97.3	89.3
	*CoNoto*								NA	NA
Gobi	*YNoto*									NA
	*GyNoto*									NA
	*CoNoto*									NA

Average percentages are indicated for the *YNoto*, *GyNoto* and *CoNoto* families (first, second and third lines for each clade). Intra-group percentages are in bold font. For the *Gypsy* TEs, the means of identity percentages are indicated for the RT/RH and the Int portions. Trem = Trematominae, Pleu = Pleuragramminae, Noto = Nototheniinae, Diss = Dissostichinae, Chan = Channichthyinae, Cygn = Cygnodraconinae, Gymn = Gymnodraconina, Arte = Artedidraconinae, Gobi = Gobionototheniinae. NA: not applicable (only one specimen per species or sequences removed because of alignment difficulties).

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
