# Peer review of "Insertion Hot Spots of DIRS1 Retrotransposon and Chromosomal Diversifications among the Antarctic Teleosts Nototheniidae"

_ijms, 2019, doi:10.3390/ijms20030701_

Round 1

Reviewer 1 Report

This is an extensive and excellent molecular cytogenetic study on transposons in a unique group of teleost fish that are not easy to get for chromosome preparations but are the more important for evolutionary biology generally and specifically for chromosomal evolution considerations.

The authors present well working FISH (no unspecific background signals present) with TEs documenting 2 types of their distribution on chromosomes, which is already an important result. Unfortunately, authors used only a single-colour FISH localizing thus merely 1 TE per FISH experiment. They neither utilized the possibility of the sequential FISH of different probes consecutively on the same metaphase, which could otherwise compensate for the single-colour regime. This is kind of pity because they missed the added value of mutual localization of these elements. 

The authors managed to show the importance of TEs for nototheniid chromosomal evolution and established a solid foundation for further studies. 

Would authors consider showing the sequence alignments used for Fig.2 e.g. in supplementary material?

Wouldnt including TE sequences of N.coriiceps from the Fish TE database (http://www.fishtedb.org/  ) improve your NJ trees? Why did you use unrooted trees?

It is unlcear what was the exact size of fragments used for FISH probes (??), however, I would refrain from drawing any conclusions on TEs copy number based on FISH with fragments shorter than 5kb as given on line 387 because of treshold limits, etc. 

l.234-235: The second part of the 1st sentence in Discussion is unclear.

In the paragraph "Fish specimens", italics is missing in species and genera names.

Technically, in my pdf printout the Fig. 2 is unreadable and Fig. 4 a bit blurred, but still well readable. 

Author Response

This is an extensive and excellent molecular cytogenetic study on transposons in a unique group of teleost fish that are not easy to get for chromosome preparations but are the more important for evolutionary biology generally and specifically for chromosomal evolution considerations.

Point 1: The authors present well working FISH (no unspecific background signals present) with TEs documenting 2 types of their distribution on chromosomes, which is already an important result. Unfortunately, authors used only a single-colour FISH localizing thus merely 1 TE per FISH experiment. They neither utilized the possibility of the sequential FISH of different probes consecutively on the same metaphase, which could otherwise compensate for the single-colour regime. This is kind of pity because they missed the added value of mutual localization of these elements. 

We agree that it would be very interesting to co-localize signals provided by several TEs. However, we have already tried to co-hybridize DIRS1and GypsyTE probes, but the DIRS1 pattern of accumulation is so strong (very shiny hot spots of insertions, see Fig 3) that it alters (by covering) the Gypsylocation. In addition, the FISH signals of those retrotransoposons are so repeated and/or accumulated that it would be technically very difficult to de-hybridize one TE and re-hybridize another TE without interference between the two. Last, the chromosomes are very sensitive to denaturation; we could not do it more than twice.

Moreover, we used fragmented cloned and Nick Translation labeled TE fragments. The same cloning vector was used for each probe. When hybridizing a single colored TE probe, vector fragments located just next to the extremity of the insert can help amplifying the TE signal. However, during the hybridization of two or more TEs, there are risks to hybridize vectors fragments between each other, thus leading to an aspecific, imprecise and undesirable background signal.

Point 2: The authors managed to show the importance of TEs for nototheniid chromosomal evolution and established a solid foundation for further studies. 

Would authors consider showing the sequence alignments used for Fig.2 e.g. in supplementary material?

Yes, absolutely. The fasta alignments have been added as Figure S1 (mentioned Line 277-278 in the legend of Figure 2). They contain all the TE sequences that we described in this analyze in addition to the DIRS1and Gypsysequences originating from the N. coriiceps genome sequencing provided by the fishTEDB (see next comment).

Wouldnt including TE sequences of N.coriiceps from the Fish TE database (http://www.fishtedb.org/  ) improve your NJ trees? Why did you use unrooted trees?

Yes, adding TE sequences of N. coriiceps from the Fish TE database would improve the trees. Unfortunately, due to the poor assembly of this genome, among the 15 DIRS1sequences and the 30 Gypsysequences available, only two sequences of DIRS1(clustering with YNotoJand R) and four sequences of Gypsy could reliably be aligned to our TEs (spanning only the RT-RH and/or Int regions) See results description Lines 250 to 254: “The two additional DIRS1sequences (766 and 1418 available in the FishTEDB [66]) originating from the genome sequencing of N. coriiceps[67] clustered with the YNotoJand theYNotoRfamilies respectively (arrows in Figure 2a). The three additional sequences of Gypsy did not clustered with a GyNotofamily identified in our analysis. The Gypsy sequence 864 was more similar to the GyNotoBand D, while sequences 99, 296 and 490 appeared to be new families (arrows in Figures 2c and 2d).” See also the revised Fig2 (additional N. coriicepsTE sequences were indicated with arrows), and the alignments presented in the FigureS1. No Copiasequence is available for N. coriicepsin the fishTEDB. 

It was possible to root our trees for the RT-RH regions of Gypsyelements with the RT-RH regions of DIRS1 TEs and reciprocally, but it was not possible for theGypsyINT regions. As we were looking for proximity between nucleotide sequences (clustering), we chose not to root our trees; unrooted trees are often used with transposable elements (Capy et al., 1982, Amrani et al., 2002, Elliot et al., 2013 for example). 

Point 3: It is unclear what was the exact size of fragments used for FISH probes (??), however, I would refrain from drawing any conclusions on TEs copy number based on FISH with fragments shorter than 5kb as given on line 387 because of treshold limits, etc. 

The exact size of the fragments used for FISH are indicated in Table S3 (insert size between 0.93 and 1.37 kb). The partial TEs previously hybridized in nototheniid species were even shorter (500bp in Ozouf-Costaz et al., 2004 and Capriglione et al., 2002), and authors mentioned “relative abundance”.

We don’t see any reference to TE copy number line 387 of the submitted version. 

Line 434 (revised version), we replaced “relative number” by “relative abundance”.

Line 516 (revised version), we replaced “high copy number” by “high abundance”.

Point 4: l.234-235: The second part of the 1st sentence in Discussion is unclear.

The sentence has been modified. Line 379-380: “The TE families YNotoJ, V, Rand BGyNotoA, B, E, F, H, I andRT, and CoNotoBidentified in the Trematomusgenomes [20] were also widely distributed in the other examined nototheniid species (Table I).”

Point 5: In the paragraph "Fish specimens", italics is missing in species and genera names.

Italics was applied to the names of the species.

Point 6: Technically, in my pdf printout the Fig. 2 is unreadable and Fig. 4 a bit blurred, but still well readable. 

The fonts of the TE families have been enlarged in Fig 2 and modified to be sharper in Fig 4.

Reviewer 2 Report

The manuscript “Insertion hot spots of DIRS1 retrotransposon and chromosomal diversifications among the Antarctic teleosts Nototheniidae” is devoted to analysis of the actual question on the role and significance of TEs during genomic evolution and speciation. The object of the study is also very perspective. They analyzed TE distribution in Antarctic fishes; teleosts Nototheniidae represent 77% of the species diversity and 91% of the fish biomass of the Southern Ocean. Furthermore, in their history there are several phase of isolation followed with fusion of the habitats in series of glacial and warming periods associated to dynamics of the icecap. However, description of material, methods, and obtained results must be improved. In Discussion the possible role of interspecific hybridization for TE burst should be included.  

Remarks:

1.       The Authors wrote (Lines 99-103): “TE families which were identified in Trematomus species [20]: YNotoJ, V, R and B, GyNotoA, B, E, F, H, I and RT, and CoNotoB were found to have an ubiquitous distribution in every examined nototheniid species (Table I). The targeted search for these elements resulted in the occasional identification of additional TE families. However, investigation of their presence and distribution in the genomes of the other species was out of the scope of the present work”.

If it is so why so many data on this point were included in manuscript.

2.       In Table 1 distribution of some TE regions (RT/RH/Int) in genomes of studied species are shown but what is “RT/RH/Int” was explained only on the line 404.  These abbreviations were wide used in the manuscript before their explanation.

3.       Author wrote in the manuscript about TE distribution in genomes of studied species. However, the method applied in the study did not allow detecting all versions of analyzed TE. The authors wrote about this problem only in the Discussion. They wrote (lines 257-261) “However, we cannot yet conclude that the lack of detection of a given TE or family in a genome (in Table 1) means they are absent. For example, no Copia TEs have been found in the genome of C. hamatus by our targeted approach (cloning of PCR products amplified with specific designed primers for each identified family), but CoNotoA and CoNotoB have been visualized on chromosomes of the crocodile icefish by heterologous FISH mapping”.

It should be explained much earlier.

4.       On the Lines 116-117 they wrote “Species in which YNoto, GyNoto, and CoNoto TEs have been located on chromosomes are presented in bold font. data from Auvinet et al. (2018) [20]”.

Are these data from Auvinet et al. (2018) or original data?

Wrong stile of citation.

5.     Lines 121-124: “When taking into account every examined nototheniid groups, this high conservation corresponds to 91% average of nucleotide identity for YNoto families, 93% or 94% for the fragments of GyNoto that span the RT/RH or the Int portions, and 96.5% for CoNoto families (Table 2)”.

I should note that authors analyzed only conservative parts of TE. They wrote about the RT/RH and the Int portions of TE but they should remind what these regions are. As I wrote above, “RT/RH/Int” was explained only on the line 404.  

6.     Lines 129-137. The giant title of the Table 2 is not applicable.

7.     Karyotype description like 2n = 47/48 should be explained.

8.     Lines 166-171: “Two major types of TE distribution patterns were identified (Figure 3A). Distribution pattern (1) is characterized by dense accumulation (hot spots of insertions) mainly in centromeric and/or pericentromeric regions, and sometimes in intercalary or telomeric positions. Distribution pattern (2) is defined by scattered, punctuate staining along chromosome arms. Combinations of these two patterns were also observed. The distributions (1) and/or (2) clearly depended upon the TE super family, but not upon TE families [20]”.

I suppose that with the applied technique it is possible to distinct Distribution pattern (1) from combinations of two described patterns. Very bright FISH signal from large cluster of TE copies could mask TE copies scattered along chromosome arms.

The used technique cannot also allow localizing TE copies precisely at centromeric or telomeric regions.

9.       Legend to Figure 3 (lines 182-183 “Red arrows point the heteromorphic Y sex chromosome in C. hamatus and the largest sub-metacentric pair in H. velifer”.  

According to Figure 3A e-h,  H. velifer karyotype contains two Y chromosomes. Is it true?

10.   Lines 184-186 “The GyNotoA, CoNotoA and CoNotoB elements were mostly dispersed throughout nototheniid chromosome arms, giving a type 2 pattern. They formed multiple spots scattered all along the chromosomes including the centromeres and telomeres”.

The used technique cannot also allow localizing TE copies exactly at centromeric or telomeric regions.

11.   Lines 196-199 “It includes also the three Trematomus species the spotted notothen T. nicolai (2n = 57/58), the emerald rockcod T. bernacchii (2n = 48), and the blunt scalyhead T. eulepidotus (2n = 24) (Ozouf et al., 1991, Morescalchi et al., 1992, 1996).”

Wrong stile of citation. T. nicolai karyotype (2n = 57/58) requires explanation.

12.   Lines 243-247: ”This conservation suggests shared TE mobilization(s) and diversification(s) occurring before the episodes of rapid speciations in nototheniid lineages (23,9 My, Matschiner et al. (2011) [70]; 22,4 My, Near et al. (2012) [16]; 15 My, Colombo et al. (2015) [19]). Comparisons between YNoto or GyNoto sequences and TEs identified in genomes of various Eupercaria species [71] indicate proximity with some copies”.

More detail description and discussion is required.

13.   Lines 262-263: “Even if FISH is not precise enough to estimate accurate TE copy numbers, we could estimate their relative number by the intensity of signals”.

Correct comparison of the signal intensity based on obtained results is impossible.

14.   Line 271. “A role of the DIRS1 in the chromosomal fusions within the family Nototheniidae?”

For discussion of mechanisms of chromosome fusion, it would be useful to analyze the distribution of telomeric repeat clusters in chromosomes of studied species.

15.   Lines 364-365 ”Fish specimens and their tissue and chromosome preparations are referenced in Table S2, with corresponding campaigns and localities of capture”.

No available data on the specimen number or the localities of capture was included in Material or Table S2.

16.   Description of Methods (Chromosome preparations, TE amplification in nototheniid genomes,  FISH with TE probes, to each chromosome preparation [20], Image acquisition and karyotyping) should be rewritten.

17.   Conclusion should be rewritten closer to obtained results.

Author Response

The manuscript “Insertion hot spots of DIRS1 retrotransposon and chromosomal diversifications among the Antarctic teleosts Nototheniidae” is devoted to analysis of the actual question on the role and significance of TEs during genomic evolution and speciation. The object of the study is also very perspective. They analyzed TE distribution in Antarctic fishes; teleosts Nototheniidae represent 77% of the species diversity and 91% of the fish biomass of the Southern Ocean. Furthermore, in their history there are several phase of isolation followed with fusion of the habitats in series of glacial and warming periods associated to dynamics of the icecap. However, description of material, methods, and obtained results must be improved. In Discussion the possible role of interspecific hybridization for TE burst should be included. 

Line 389 to 390: timing of the TE mobilization and diversification into nototheniid genome given their conservation. We added Line 418 to 421: “Since speciation is not a discrete process and reproductive barriers take time to establish (evolutionary scale of time), we can not exclude the influence of interspecific hybridizations in favoring transpositions as described in plants or insects in a genomic destabilization and re-organization context [75].” See comment 12 for further precisions.

Remarks:

1.       The Authors wrote (Lines 99-103): “TE families which were identified in Trematomus species [20]: YNotoJ, V, R and BGyNotoA, B, E, F, H, I and RT, and CoNotoB were found to have an ubiquitous distribution in every examined nototheniid species (Table I). The targeted search for these elements resulted in the occasional identification of additional TE families. However, investigation of their presence and distribution in the genomes of the other species was out of the scope of the present work”.

If it is so why so many data on this point were included in manuscript.

The targeted search using our specific primers amplified TE families correctly, resulting in a wide prospection and distribution among the examined nototheniid species. We did not further investigate the new TE families identified by clustering. However we included them in our TE analysis. This inclusion makes the overview of the elements present in the genomes more complete. Pointing out their presence can serve as a foundation for future work instead of keeping them unpublished and unshared with the scientific community.

We modified the text to clarify this. Line 122 to 124: “While investigation of their presence and distribution in the genomes of the other species was out of the scope of the present work, we included them into the analysis to complete the overview of TEs in Nototheniid genomes”.

2.       In Table 1 distribution of some TE regions (RT/RH/Int) in genomes of studied species are shown but what is “RT/RH/Int” was explained only on the line 404.  These abbreviations were wide used in the manuscript before their explanation.

Explained earlier Line 120 to 121: “The targeted search for these elements spanning the Reverse Transcriptase/RNAseH (RT-RH) and/or the Integrase (Int) conserved domains resulted in the occasional identification of additional TE families.” and also in the legend of Table 1 Line 144: “RT/RH stands for the Reverse Transcriptase/RNAseH and Int for Integrase conserved domains.”

3.       Author wrote in the manuscript about TE distribution in genomes of studied species. However, the method applied in the study did not allow detecting all versions of analyzed TE. The authors wrote about this problem only in the Discussion. They wrote (lines 257-261) “However, we cannot yet conclude that the lack of detection of a given TE or family in a genome (in Table 1) means they are absent. For example, no Copia TEs have been found in the genome of C. hamatus by our targeted approach (cloning of PCR products amplified with specific designed primers for each identified family), but CoNotoA and CoNotoB have been visualized on chromosomes of the crocodile icefish by heterologous FISH mapping”.

It should be explained much earlier.

We add an explanation earlier Line 131 to 133: “However, as amplification of cloned PCR products was used in this study, we assume that a lack of identification of a given TE family in a species using this approach does not mean they are absent from their genome.” It is also mentioned in the legend of Table1, Line 127-128 : “- : TEs tested but not found using our approach.”

4.       On the Lines 116-117 they wrote “Species in which YNotoGyNoto, and CoNoto TEs have been located on chromosomes are presented in bold font. data from Auvinet et al. (2018) [20]”.

Are these data from Auvinet et al. (2018) or original data?

There are original data, except for the species labeled with *. The * has been forgotten in the legend, but has been added. It should read Line 146: “*: data from Auvinet et al. [20] “.

Wrong stile of citation. Modified

5.     Lines 121-124: “When taking into account every examined nototheniid groups, this high conservation corresponds to 91% average of nucleotide identity for YNoto families, 93% or 94% for the fragments of GyNoto that span the RT/RH or the Int portions, and 96.5for CoNoto families (Table 2)”.

I should note that authors analyzed only conservative parts of TE. They wrote about the RT/RH and the Int portions of TE but they should remind what these regions are. As I wrote above, “RT/RH/Int” was explained only on the line 404.  

We agree that we worked on conserved domains, but even for conserved regions, these nucleotide sequence identity represent a high degree of conservation. 

“RT-RH and Int” explained earlier, see answer to comment 3.

6.     Lines 129-137. The giant title of the Table 2 is not applicable.

It has been changed Line 160 for: “TE intrafamily conservation within and across nototheniid clades.”

7.     Karyotype description like 2n = 47/48 should be explained.

Modified for: 2n=47(male)/48(female) for C. hamatusLine 287 and for 2n=57(male)/58(female) Line 333 for T. nicolai(the first time they appear in the text).

8.     Lines 166-171: “Two major types of TE distribution patterns were identified (Figure 3A). Distribution pattern (1) is characterized by dense accumulation (hot spots of insertions) mainly in centromeric and/or pericentromeric regions, and sometimes in intercalary or telomeric positions. Distribution pattern (2) is defined by scattered, punctuate staining along chromosome arms. Combinations of these two patterns were also observed. The distributions (1) and/or (2) clearly depended upon the TE super family, but not upon TE families [20]”.

I suppose that with the applied technique it is possible to distinct Distribution pattern (1) from combinations of two described patterns. Very bright FISH signal from large cluster of TE copies could mask TE copies scattered along chromosome arms.

There could be some masking. However the signals produced by a given TE in some species do not necessarily overlap. In our results, we see a mix between patterns 1 and 2 in the cases where DIRS1forms rare scattered signals in addition to hot spots of insertion (Fig 3B b) and where Copia Hydraforms minor accumulations in addition to the majority of scattered spots (Fig3Ad and l).

We agree that very bright signals could mask a part of the little dispersed spots and that is one of the reasons why we did not co-hybridize several TEs together on nototheniid chromosomes (see first answer to reviewer 1).

The used technique cannot also allow localizing TE copies precisely at centromeric or telomeric regions.

The centromeric and telomeric regions are not very precisely delimited at this scale of observation. Hence the use of “near the centromere” Line 327. 

We modified Line 28: “(mainly next to pericentromeric and centromeric regions).”

Sentence modified Line 292 to 293: “Two major types of TE distribution patterns were identified (Figure 3A). Distribution pattern (1) is characterized by dense accumulation (hot spots of insertions) mainly next to centromeric and/or pericentromeric regions, and sometimes in intercalary or near telomeric positions.”

Centromeres and telomeres modified in Line 316 for “next to the centromeric and telomeric regions.” 

Lines 484-485 also modified: “…the accumulation of numerous copies next to pericentromeric and centromeric regions in almost every chromosomal pairs as well as near the telomeric regions for some of them”.

Line 503 also modified: “can rarely form very little spots next to centromeric regions.”

9.       Legend to Figure 3 (lines 182-183 “Red arrows point the heteromorphic Y sex chromosome in C. hamatus and the largest sub-metacentric pair in H. velifer”.  

According to Figure 3A e-h,  H. velifer karyotype contains two Y chromosomes. Is it true?

No, this might have been unclear. We replaced the red arrows by orange arrows for H. velifer. The red arrows show the Y sex chromosome in C. hamatus, whereas the purple arrows inH. velifershow another chromosomal pair (autosomes not linked with sex differenciation) recognizable thanks to it large size and it sub-metacentric characteristics. Legend of Fig3 Line 312 and text Lines 319 and 341 were changed as well.

10.   Lines 184-186 “The GyNotoACoNotoA and CoNotoB elements were mostly dispersed throughout nototheniid chromosome arms, giving a type 2 pattern. They formed multiple spots scattered all along the chromosomes including the centromeres and telomeres”.

The used technique cannot also allow localizing TE copies exactly at centromeric or telomeric regions.

Centromeres and telomeres modified in Lines 292, 316, 484 and 503 for “next to the centromeric and telomeric regions.” See comment 8.

11.   Lines 196-199 “It includes also the three Trematomus species the spotted notothen T. nicolai (2n = 57/58), the emerald rockcod T. bernacchii (2n = 48), and the blunt scalyhead T. eulepidotus (2n = 24) (Ozouf et al., 1991, Morescalchi et al., 1992, 1996).”

Wrong stile of citation. T. nicolai karyotype (2n = 57/58) requires explanation.

Citation corrected. Karyotype of T. nicolaicorrected for 57(male)/58(female) Line 333, see comment 7.

12.   Lines 243-247: ”This conservation suggests shared TE mobilization(s) and diversification(s) occurring before the episodes of rapid speciations in nototheniid lineages (23,9 My, Matschiner et al. (2011) [70]; 22,4 My, Near et al. (2012) [16]; 15 My, Colombo et al. (2015) [19]). Comparisons between YNoto or GyNoto sequences and TEs identified in genomes of various Eupercaria species [71] indicate proximity with some copies”.

More detail description and discussion is required.

We propose adding Line 410-421: “Further alignments with TEs identified in other Eupercaria species [74] could give information about the timing of TE mobilization and diversification in those genomes. Preliminary results indicate proximity with some copies. These shared TEs would indicate their acquisitions in a common ancestor, before the separation between Antarctic and temperate species and the formation of the Antarctic convergence. However, better additional alignments with well assembled Eupercaria TE sequences and non-chimeric elements would be needed to further investigate the origin of the YNotoGyNoto and CoNotoin nototheniid genomes and to relate it to the geographic and oceanographic events leading to the isolation of this teleost group from temperate species. Since speciation is not a discrete process and reproductive barriers take time to establish (evolutionary scale), we can not exclude the influence of interspecific hybridizations in favoring transpositions as described in plants or insects in a genomic destabilization and re-organization context [75].”

13.   Lines 262-263: “Even if FISH is not precise enough to estimate accurate TE copy numbers, we could estimate their relative number by the intensity of signals”.

Correct comparison of the signal intensity based on obtained results is impossible.

We have removed “we could estimate their relative number by the intensity of signals” and replaced it Line 433-434 by “Even if FISH is not precise enough to estimate accurate TE copy numbers, we could compare the relative abundance between signals provided by different TEs”.

14.   Line 271. “A role of the DIRS1 in the chromosomal fusions within the family Nototheniidae?” 

For discussion of mechanisms of chromosome fusion, it would be useful to analyze the distribution of telomeric repeat clusters in chromosomes of studied species.

Accumulations near telomeric regions are included in the distribution pattern (1) (e.g Line 292). Moreover, DIRS1accumulate mostly in peri-centromeric regions, and fusions are mostly centric fusions in the Nototheniidae. Distribution of telomeric repeat cluster give information on the possible tandem fusions, which appeared to be very punctual in this group of species (mechanism described Line 475). We modified Line 482-483:”Such an environmental instability may have boosted DIRS1mobilizations in nototheniid genomes and the accumulation of numerous copies next to pericentromeric and centromeric regions in almost every chromosomal pairs as well as near the telomeric regions for some of them.”

15.   Lines 364-365 ”Fish specimens and their tissue and chromosome preparations are referenced in Table S2, with corresponding campaigns and localities of capture”.

No available data on the specimen number or the localities of capture was included in Material or Table S2. 

Localities of captures were added in table S2.

16.   Description of Methods (Chromosome preparations, TE amplification in nototheniid genomes,  FISH with TE probes, to each chromosome preparation [20], Image acquisition and karyotyping) should be rewritten.

Chromosomal preparation, FISH with TE probes and image acquisition have been modified and re-written, as well as the TE amplification section.

17.   Conclusion should be rewritten closer to obtained results.

Conclusion has been rewritten to take into account all the obtained results.

Round 2

Reviewer 2 Report

The authors answer the questions and made a lot of corrections and modification in the text of manuscript/

Nevertheless, I have to make one small remark and ask for two additional explanations.

Remark 1. Earlier I have written, “In Table 1 distribution of some TE regions (RT/RH/Int) in genomes of studied species are shown but what is “RT/RH/Int” was explained only on the line 404.  These abbreviations were wide used in the manuscript before their explanation”.

The answer was “Explained earlier Line 120 to 121: “The targeted search for these elements spanning the Reverse Transcriptase/RNAseH (RT-RH) and/or the Integrase (Int) conserved domains resulted in the occasional identification of additional TE families.” and also in the legend of Table 1 Line 144: “RT/RH stands for the Reverse Transcriptase/RNAseH and Int for Integrase conserved domains.”

I have to note that abbreviation “RT-RH” differs from the “RT/RH”. It is better to correct.

Remark 2. In Figure 4, the chromosome from C. hamatus contains strong and large FISH signal on one of chromatids and no FISH signal at this site on the second one. How is it possible? What means the underlining of the number on this Figure (C. hamatus (2n=47/48))? It is necessary to explain in the legend.

Author Response

Author's Notes to Reviewer were attached below.
